# SemVAD: Fusing Semantic and Vision Features for Weakly Supervised Video Anomaly Detection

**Hamza Karim**                                                                                     *hamzakarim@usf.edu*
*Department of Electrical Engineering*
*University of South Florida*

**Yasin Yilmaz**                                                                                     *yasiny@usf.edu*
*Department of Electrical Engineering*
*University of South Florida*

**Reviewed on OpenReview:** *https://openreview.net/forum?id=6tkvxrHidI&referrer*

## Abstract

In recent years, vision-language models such as CLIP and VideoLLaMA have demonstrated the ability to express visual data in semantically rich textual representations, making them highly effective for downstream tasks. Given their cross-modal semantic representation power, leveraging such models for video anomaly detection (VAD) holds significant promise. In this work, we introduce Semantic VAD (SemVAD), a novel methodology for weakly supervised video anomaly detection (wVAD) that effectively fuses visual and semantic features obtained from pretrained vision-language models, specifically VideoLLaMA 3 and CLIP. Our approach enhances performance and explainability in anomaly detection. Additionally, we analyze the sensitivity of recent state-of-the-art models to randomness in training initialization and introduce a more comprehensive evaluation framework to assess their robustness to small changes in training such as the seed of random number generator. This framework aims to provide a more rigorous and holistic assessment of model performance, ensuring a deeper understanding of their reliability and reproducibility in wVAD.

## 1 Introduction

In recent years, weakly supervised anomaly detection (wVAD) has gained increasing attention due to its potential applications in automated surveillance and content moderation systems. The majority of existing research has focused on contrastive learning-based approaches, where video content is segmented and transformed into visual embeddings, followed by training a neural network using multiple-instance learning (MIL) (Sultani et al., 2018). Some recent studies have explored alternative strategies, including pseudo-labeling instead of MIL (Karim et al., 2024). However, MIL-based methods have remained the dominant approach in the field, demonstrating their effectiveness in handling weakly labeled video anomaly detection tasks.

MIL-based methods exhibit a sensitivity drawback, where model convergence is highly dependent on weight initialization. This issue arises from the inherent characteristics of the MIL framework, namely the random selection of video clips . Consequently, model performance may vary considerably depending on the choice of the random seed.

Recently, pretrained vision-language models (VLMs) have gained significant attention due to their ability to learn rich visual representations with semantic understanding (Radford et al., 2021), (Zhang et al., 2023), (Cheng et al., 2024), (Zhang et al., 2025). CLIP(Radford et al., 2021), has gained significance in recent wVAD literature (Wu et al., 2024)(Joo et al., 2023) and has been primarily used as backbone for feature extraction due to its strong visual representation. Although CLIP is trained on image-text pairs, researchers have introduced feature encoding techniques to incorporate temporal dependencies, such as Temporal Self-

Attention (TSA) (Pu et al., 2024) and the Local-Global Temporal Adapter (LGT-Adapter) (Wu et al., 2024).

However, recent advancements in vision-language models have led to the development of video-language models that can directly process and understand video content. In this study, we aim to adopt the MIL framework in conjunction with VLMs to generate descriptive annotations for anomalies occurring in video streams and utilize these descriptions to enhance the performance and explainability in wVAD. Our key contributions are as follows:

1. We propose *Semantic VAD (SemVAD)* featuring a novel feature fusion architecture that integrates semantic and visual features extracted by video-language models (Section 3), which distinguishes our method from the existing methods that use LLM/VLM for unsupervised VAD (Yang et al., 2024; Zanella et al., 2024) or weakly supervised (Pu et al., 2024; Joo et al., 2023; Lv et al., 2023; Wu et al., 2024) VAD.

2. We introduce an evaluation criterion for assessing the robustness of wVAD systems to randomness in training initialization (Section 4.2).

3. We demonstrate improved performance and explainability over recent state-of-the-art methods in both coarse-grained and fine-grained anomaly detection tasks (Sections 4.4, 4.5).

## 2 Related Work

### 2.1 Weakly supervised video anomaly detection

Sultani et al. (Sultani et al., 2018) introduced the Multiple Instance Learning (MIL) framework, laying the groundwork for weakly supervised video anomaly detection (wVAD). MIST followed the MIL framework and used an encoder-based method that fine-tunes a feature encoder based on the generated pseudo-labels. Subsequent work improved performance via temporal aggregation. RTFM (Tian et al., 2021) proposed a Multi-Scale-Temporal Network (MTN) to better aggregate temporal features. Later methods like MGFN (Chen et al., 2023) and S3R (Wu et al., 2022) incorporated dictionary and feature magnitude learning to advance wVAD. Although alternative approaches have been explored, such as the use of k-NN distances to generate pseudo-labels (Karim et al., 2024), MIL remains a prevalent choice due to the effectiveness of its aggregation techniques.

Recently, large language models have motivated LLM based anomaly detection, especially in unsupervised VAD, focusing on model reasoning (Yang et al., 2024) and training-free approaches. LAVAD (Zanella et al., 2024) proposes a training-free, fully unsupervised pipeline that converts video into language and lets an LLM do the temporal reasoning. Frames are captioned by a VLM; captions are cleaned with cross-modal similarity; then prompting aggregates temporal evidence and yields anomaly scores—no fine-tuning or domain data required. While the strengths are simplicity and zero annotation cost, limitations include sensitivity to caption noise and reliance on general-purpose captioners/LLMs, which can miss subtle anomalies or bias toward "textually salient" events. Holmes-VAD (Zhang et al., 2024) is another LLM based model that proposed to train a Multi-Modal LLM on VAD-Instruct50k, a large instruction-tuning corpus built with semi-automatic single-frame annotations, through human effort and LLM-generated explanations.

Prompt-learning remain popular in wVAD (Pu et al., 2024; Joo et al., 2023; Lv et al., 2023; Wu et al., 2024). CLIP-TSA (Joo et al., 2023) used CLIP features and temporal self-attention to model short- and long-term dependencies. UMIL (Lv et al., 2023) introduced an Unbiased MIL framework leveraging CLIP features to reduce bias and improve wVAD. Pel4VAD(Pu et al., 2024) introduced prompt learning to discriminate between nominal and anomalous segments. VADClip (Wu et al., 2024) recently introduced the Local-Global Temporal Adapter (LGT-Adapter), combining windowed self-attention with lightweight GCNs to capture fine-grained and long-range temporal dependencies, significantly boosting detection performance.

None of the existing LLM/VLM-based methods for unsupervised VAD (Yang et al., 2024; Zanella et al., 2024) or weakly supervised (Pu et al., 2024; Joo et al., 2023; Lv et al., 2023; Wu et al., 2024) VAD fuses visual and semantic features from videos like SemVAD.

## 2.2 Vision language models

VLMs have become central to bridging visual understanding with natural language reasoning. Among the most foundational works is CLIP (Radford et al., 2021), which introduced a contrastive learning framework that aligns images and text in a shared embedding space. Trained on hundreds of millions of image-text pairs, CLIP enables zero-shot transfer by embedding both modalities using separate encoders and learning to maximize their similarity for matching pairs. Its generalizability across various downstream tasks, such as image retrieval, classification, and video understanding, has made it a cornerstone of multi-modal learning and a powerful feature extractor in weakly supervised setups.

More recent models have expanded this paradigm into temporally complex and generative domains. Sora (Liu et al., 2024) shifts the focus from understanding to generation by producing high-fidelity videos directly from text prompts. Through diffusion-based modeling, Sora can generate coherent and realistic video sequences, marking a leap in cross-modal synthesis and showcasing how language can condition fine-grained spatiotemporal outputs. Meanwhile, Video-LLaMA (Zhang et al., 2023),(Zhang et al., 2025) integrates large language models with video encoders to enable multimodal reasoning over sequential visual data. It builds on instruction-tuned LLMs and pretrained vision backbones to handle tasks like video question answering and temporal captioning, combining perception with structured language reasoning.

# 3 Method

## 3.1 Problem Definition and Method Overview

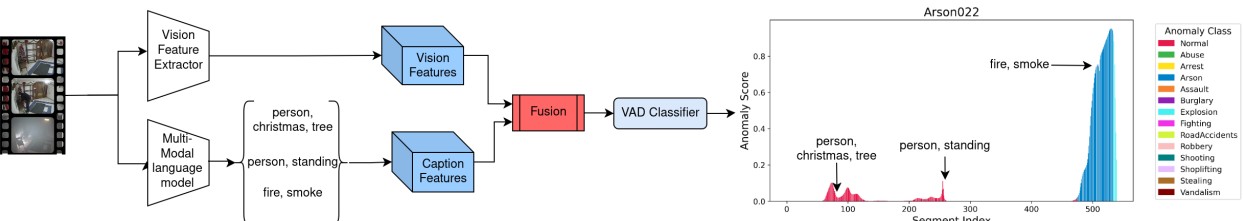

Figure 1: Overview of the SemVAD architecture, which leverages captions generated by a multi-modal language model through fusion with extracted vision features and outputs anomaly score for the given anomaly classes.

SemVAD introduces a video anomaly detection (VAD) architecture designed to harness the capabilities of recent multimodal language models by integrating them with established vision feature extractors such as CLIP, I3D, and Uniformer. The overall architectural pipeline of SemVAD is depicted in Figure 1.

Let $\mathcal{V}$ denote the set of all videos, where each video $V \in \mathcal{V}$ may contain either normal content or an anomaly belonging to one of $M$ predefined anomaly classes. Each video $V$ is divided into $N$ non-overlapping clips, denoted as $\{v_1, v_2, \ldots, v_N\}$. The wVAD setting assumes that only video-level labels are available during training. Specifically, given a video $V$, if all frames are free of abnormal events, the video is labeled as normal $y = 0$. Conversely, if at least one frame contains an anomaly, the video is labeled as abnormal $y = 1$. The objective of a wVAD system is to learn a model capable of predicting frame-level anomaly scores for coarse-grained (normal vs. anomalous) and fine-grained (anomaly classes) decisions, despite being trained solely on video-level annotations.

In this work, we semantically enhance the CLIP (Radford et al., 2021) (visual) features used in VADCLIP (Wu et al., 2024) by integrating a caption $C_i$ for each clip $v_i$ obtained by the multi-modal language model, VideoLLaMA 3 and postprocessing. A novel two-phase fusion module consisting of attention and convolution mixer steps is proposed for seamless integration. We further enhance the model's temporal reasoning and anomaly localization capabilities by training the model with a novel semantic-alignment loss function in addition to the Local-Global Temporal (LGT) Adapter and the dual-branch MIL-align framework of VAD-

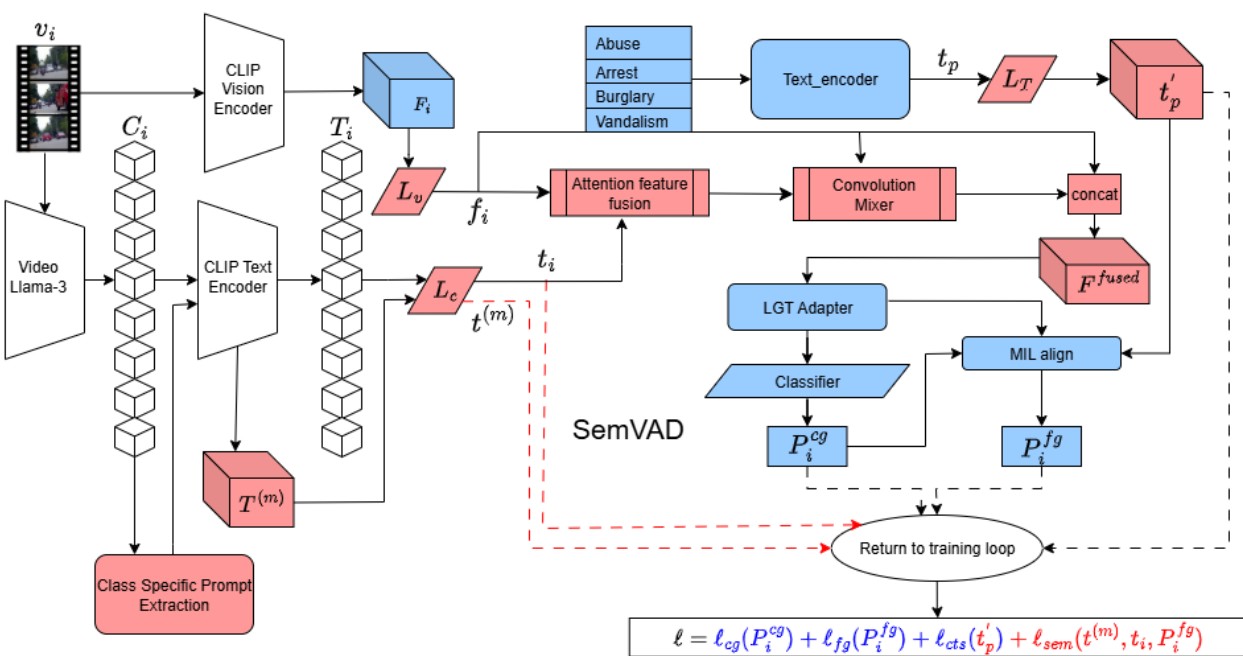

Figure 2: Overview of the SemVAD architecture, which builds upon the VADCLIP (Wu et al., 2024) framework by incorporating semantically rich captions to enhance visual representations. Descriptive captions, generated via VideoLLaMA 3, are fused with CLIP visual features through a two-phase fusion module comprising an attention mechanism and a convolutional mixer. Additionally, SemVAD introduces a novel class-specific alignment loss $\ell_{sem}(t_i, t^{(m)}), P_i^{fg})$, which encourages caption embeddings $t_i$ to align with class-specific prompts $t^{(m)}$. Components highlighted in blue represent the original VADCLIP pipeline while red modules denote the proposed contributions in SemVAD.

CLIP. A detailed overview of the proposed Semantic VAD (SemVAD) method is shown in Figure 2 with novel modules and VADCLIP modules highlighted in red and blue, respectively.

### 3.2 Extraction of semantic features using VideoLLaMA 3

For each clip $v_i$ in a video $V$, we generate a caption $C_i$ to describe the content of the clip. To incorporate temporal context, a window of three consecutive clips $v_{i-1}, v_i, v_{i+1}$ is passed to VideoLLaMA 3 with the following prompt:

> **"You are given three consecutive clips. Describe what is happening in the middle clip using the preceding and the following clip as context."**

The three-clip window is slid across the sequence of $N$ clips in the video, with padding of one clip at the beginning and end as needed. Each caption is then embedded into a numerical vector $T_i \in \mathbb{R}^Q$ using the CLIP text encoder, where $Q$ is the dimension of the feature vector. We also incorporate a system-level prompt provided in appendix A.1

During training, an important goal is to encourage captions that are likely to contain anomalies to be pushed towards their corresponding class labels. To this end, we first derive a generalized caption representation for each anomaly class. The generalized caption for each class serves as a prototype or centroid which captures the key semantic features that characterize anomalies within that class. Let $m = 0$ represent the normal class and $m = 1, \ldots, M$ represent the anomaly classes. Figure 3 shows the complete pipeline to obtain generalized caption $\mathcal{C}^{(m)}$ for each class from the generated captions $C_i$. First, the set of class-specific captions $\mathcal{C}^{(m)}$ is

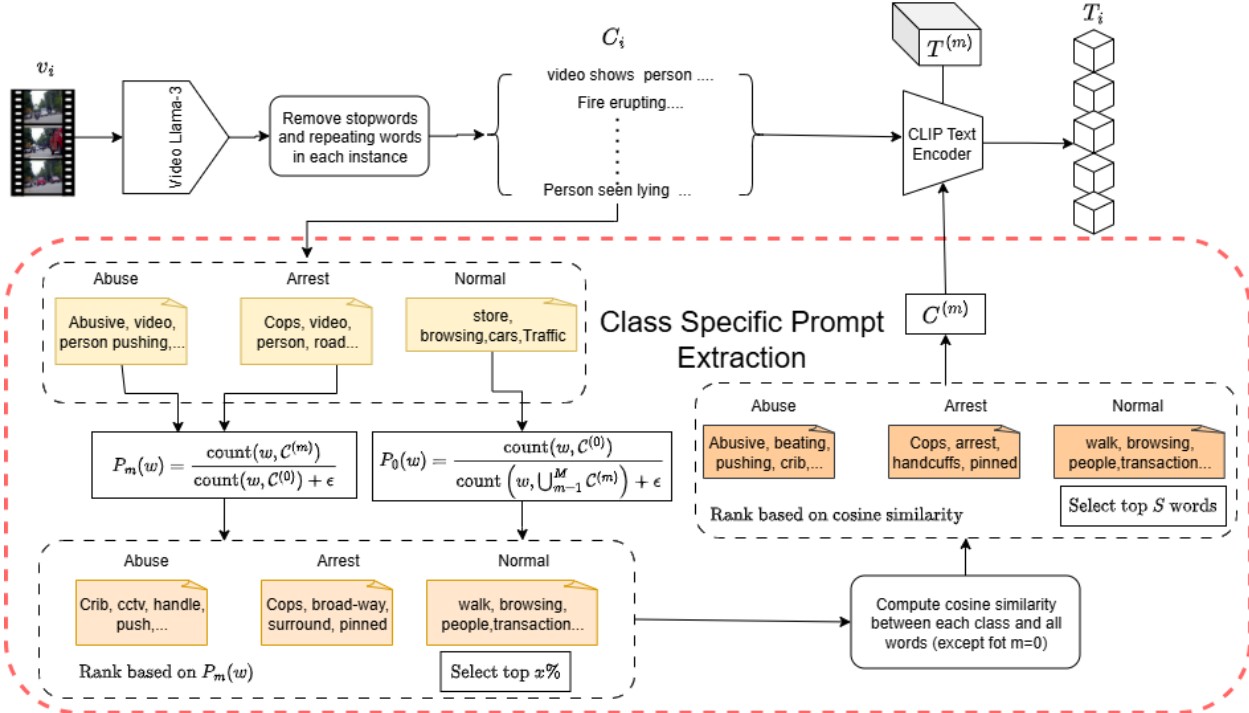

Figure 3: Class-Specific Prompt Extraction Pipeline: Captions $C_i$ are generated for each clip $v_i$ using VideoLLaMA 3 model. After filtering, words are ranked by class-specific scores $P_m(w)$, and top words are embedded via CLIP. For anomaly classes, words are re-ranked using cosine similarity; for the normal class, top words are selected directly. Final class prompts are encoded to obtain CLIP text features $T_m$.

formed from all clips that belong to videos labeled with the $m$-th anomaly class. Similarly, let $\mathcal{C}^{(0)}$ represent the set of captions from all clips in videos labeled as normal.

For each unique word $w$ appearing in the caption corpus, we define the class-specific score $P_m(w)$ as follows:

$$P_m(w) = \frac{\text{count}(w, \mathcal{C}^{(m)})}{\text{count}(w, \mathcal{C}^{(0)}) + \epsilon}, \ m > 0; \ P_0(w) = \frac{\text{count}(w, \mathcal{C}^{(0)})}{\text{count}\left(w, \bigcup_{m=1}^{M} \mathcal{C}^{(m)}\right) + \epsilon}, \tag{1}$$

where $\text{count}(w, \mathcal{C}^{(m)})$ denotes the number of times the word $w$ appears in the set of captions $\mathcal{C}^{(m)}$, and $\epsilon$ is a small constant to avoid division by zero.

For each class $m$, each word $w$ is first ranked according to its score $P_m(w)$. The top $x\%$ of words with the highest scores are selected to represent salient linguistic features for the $m$-th anomaly class. Each selected word is then embedded into a vector representation using the CLIP text encoder. These word embeddings are then compared to the CLIP-encoded representation of the $m$-th anomaly class label via cosine similarity. Based on this similarity, the words are re-ranked, and the top $S$ most semantically relevant words are selected.

For the normal class, the cosine similarity step is omitted due to the inadequacy of the phrase "normal" as a semantically meaningful or descriptive label in the CLIP embedding space. Instead, the top $S$ words are selected directly based on their $P_0(w)$ scores, following the same ranking procedure described earlier. The final set of $S$ words for each class $m$ are concatenated using a space delimiter to form the generalized caption $C^{(m)}$. This phrase is then passed through the CLIP text encoder to obtain a numerical representation $T^{(m)} \in \mathbb{R}^Q$.

### 3.3   Vision-Caption Fusion

We propose a two-phase fusion module designed to integrate caption features $T = [T_i]$ of a video with visual features $F = [F_i] \in \mathbb{R}^{N \times D}$ obtained from the CLIP visual encoder. The fusion process begins by aggregating caption features globally through an attention mechanism conditioned on the visual features. This is followed by a convolution mixer module that fuses the attended features along the local temporal dimension. This is similar to local-global aggregation in WSVAD Wu et al. (2022),Wu et al. (2024),Chen et al. (2023).

#### 3.3.1   Attention-Based Fusion

In the first phase, as illustrated in Figure 4, the CLIP visual features $F$ and the caption features $T$ for a video are passed through separate learnable linear projection layers, denoted as $L_v$ and $L_c$, to obtain projected visual and textual features, $f \in \mathbb{R}^{N \times D}$ and $t \in \mathbb{R}^{N \times D}$, where $D$ is the feature dimensionality and $N$ is the number of clips per video.

Next, cross-modal attention is computed between the projected visual features $f$ and textual features $t$. The resulting attention scores are used to aggregate the textual features based on their similarity with the visual modality. The aggregated caption features are then added to the visual embeddings, followed by layer normalization and a feed-forward network. This results in the attention-fused feature representation, denoted as $t^{\mathrm{attn}} \in \mathbb{R}^{N \times D}$.

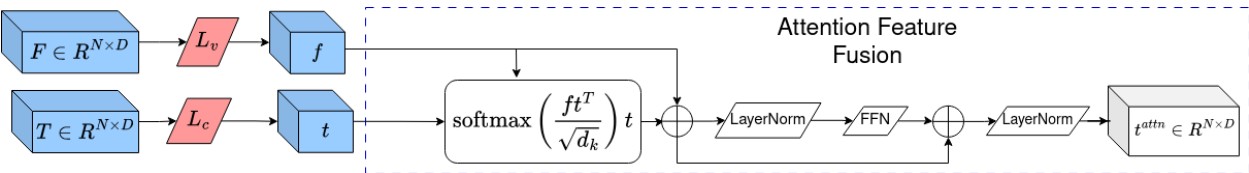

Figure 4: Illustration of the attention-based vision-caption fusion module.

In our cross-modal attention setup, we treat vision features $f$ as queries and text features $t$ as keys. Rather than employing the standard QKV attention, we found that reusing the keys as values (QKK attention) consistently yielded better performance (see Appendix A.4). While avoiding an additional projection for values, QKK seems to preserve semantic alignment, acting as a form of regularization in the fusion process.

#### 3.3.2   Convolution Mixer

Next, the attention-fused text features $t_i^{\mathrm{attn}}$ are stacked with the projected visual features $f_i$ to produce a combined representation $X^{\mathrm{stack}} = [x_i^{\mathrm{stack}}] \in \mathbb{R}^{2 \times N \times D}$, as shown in Figure 5. A 2D convolutional layer with a kernel size of $2 \times 3$ is applied to this stacked tensor, with both the input and output channels set to $D$. This convolution operation, which fuses the features across the modality (stack) and temporal dimensions, is followed by the GELU activation function. Necessary padding is applied along the temporal axis to preserve the clip length. The output of this operation is denoted as $X^{\mathrm{conv}} \in \mathbb{R}^{N \times D}$, which is subsequently passed

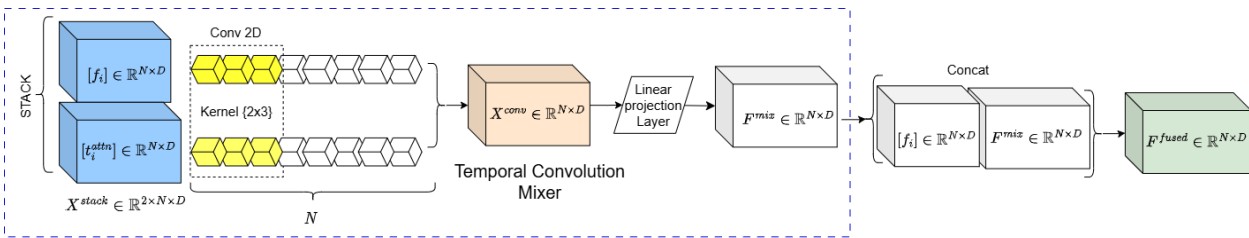

Figure 5: Illustration of the convolution mixer module.

through a linear layer to obtain the convolution-mixed features $F^{\mathrm{mix}} = [f_i^{\mathrm{mix}}] \in \mathbb{R}^{N \times D}$.

Finally, the visual features $f_i$ and $f_i^{\text{mix}}$ are concatenated along the feature dimension to produce the fused representation $F^{\text{fused}} = [f_i^{\text{fused}}] \in \mathbb{R}^{N \times 2D}$, which is then used as input to the LGT adapter. In Section 4.8, we further explore the effect of each module on the performance of our model. The fused features are then passed through the LGT temporal aggregator before computing the coarse-grained (binary: normal vs. anomaly) $P_i^{\text{cg}}$ and fine-grained (multi-class for also detecting anomaly type) $P_i^{\text{fg}}$ classification probabilities, as in (Wu et al., 2024).

### 3.4 Loss Function

We build upon the commonly used top-$K$ Multiple Instance Learning (MIL) loss, $\ell_{\text{cg}}$ (Wu et al., 2024), to supervise the learning of coarse-grained anomaly scores. In addition, we incorporate the multi-class MIL alignment loss $\ell_{\text{fg}}$ for fine-grained anomaly scores and the contrastive loss $\ell_{\text{cts}}$ for pushing normal class away from anomaly classes, both originally introduced in VADCLIP (Wu et al., 2024). These losses encourage the projected representation $t_p'$ of anomaly class labels to diverge from both the normal class embedding and embeddings of other anomaly classes A.5.

To further enhance class-specific alignment, we introduce an additional loss term that explicitly aligns the projected caption embedding $t_i$ towards its corresponding anomaly class embedding $t^{(m)}$ for top-$K$ anomalous clips, which is obtained by applying the linear caption projection layer $L_c$ to the caption embedding $T^{(m)}$. This is achieved by MIL alignment using the fine-grained (multiclass) probability scores $P_i^{\text{fg}}$, thereby reinforcing the semantic consistency between the caption and its target anomaly class.

Since we know the label $m$ for each video in the training set, in the probability matrix $[P_i^{\text{fg}}] \in [0,1]^{N \times M}$, we select the $m$-th column to get $s_m \in [0,1]^N$.

Let $\{i_1, \ldots, i_K\} = \arg\max_K \{s_m\}$ denote the indices of the top-$K$ clips with the highest probabilities in $s_m$ and $t_{i_k} \in \mathbb{R}^D, k = 1, \ldots, K$, denote the corresponding feature vectors.

Next, we compute the average cosine distance $\ell_{\text{sem}}$ between the semantic embedding $t^{(m)}$ of class $m$ and the semantic embedding $t_{i_k}$. The overall training loss is given by the sum of four loss functions:

$$\ell_{\text{sem}} = \text{avg}\left\{1 - \frac{t_{i_k}^T t^{(m)}}{\|t_{i_k}\| \|t^{(m)}\|} : k = 1, \ldots, K\right\}, \quad \ell = \ell_{\text{sem}} + \ell_{\text{cg}} + \ell_{\text{fg}} + \ell_{\text{cts}}.$$

## 4 Experiments

### 4.1 Datasets

We evaluate and compare the proposed method on two widely used datasets in video anomaly detection: UCF-Crime (Sultani et al., 2018) and XD-Violence (Wu et al., 2020). For consistency and fairness, we utilize only the visual modality and discard any accompanying audio information. Both datasets provide weakly labeled training videos, making them suitable for wVAD.

### 4.2 Evaluation Metrics

Following prior work, for coarse-grained (normal vs. anomalous) anomaly detection, we use the area under the receiver operating characteristic curve (AUC) for UCF-Crime and average precision (AP) for XD-Violence to ensure comparability with existing methods. Moreover, for fine-grained (multi-class) anomaly detection, we use mean average precision (mAP) following Wu et al. (Wu et al., 2024).

While most wVAD studies report only the "best" AUC or AP, our experiments reveal significant performance variability arising from the stochastic nature of training, particularly in contrastive loss settings like multiple instance learning (MIL). MIL-based models are highly sensitive to random initialization and sampling, often resulting in chaotically fluctuating performance (Figure 6b).

At the beginning of training, randomly initialized parameters yield arbitrary anomaly scores for video clips $\{v_i\}$. When the top-$K$ clips are selected for training supervision, there is a substantial risk that they lack true anomalies, leading to optimization toward suboptimal minima. This instability mainly stems from the random initialization of network weights at training start and the stochastic pairing of normal and anomalous segments during MIL-based contrastive learning.

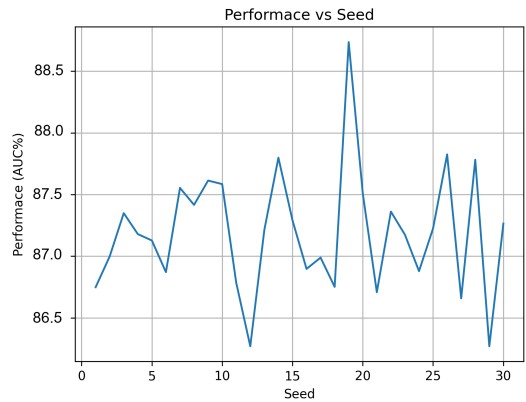

| Weight Initialization Seed | Data Seed | AUC (%) |
|---|---|---|
| 234 | 234 | 87.44 |
| 9 | 9 | 85.61 |
| 234 | 9 | 88.24 |
| 9 | 234 | 86.33 |

(a) Impact of weight initialization and data shuffling seeds on VADCLIP (Wu et al., 2024) performance (AUC) on UCF-Crime.

(b) Chaotic AUC variations of SemVAD on UCF-Crime as a function of weight initialization seed.

Figure 6: Effect of random seeds on VAD performance.

Figure 6a shows the performance variability of the state-of-the-art VADCLIP (Wu et al., 2024) under different random seeds on UCF-Crime. Performance depends not only on model weight initialization but also on the stochastic pairing of video segments (Data Seed). While the best AUC was originally reported as 88.02% in (Wu et al., 2024), our experimentation with several seed pairs reveal notable deviations and even a better AUC (88.24%) than reported. Figure 6b demonstrates the non-smooth AUC variations of our proposed SemVAD method on UCF-Crime. Wild performance variations (Figure 6a and Figure 6b) display the impracticality of optimizing random seed as if it was a hyperparameter. Hence, evaluating the performance of wVAD methods in terms of best performance as a function of random seed does not provide a fair comparison, where the best achievable performance is likely never known.

To address this, we adopt a more robust evaluation strategy by training each model with ten random seeds and reporting the mean and standard deviation of performance metrics. This provides a more reliable and reproducible assessment of model effectiveness and stability.

## 4.3 Implementation Details

We adopt the VADCLIP pipeline and utilize the CLIP (ViT-B/16) architecture as the backbone. We set the percentage $x$ of selected keywords to be 10%, as higher values do not increase performance while requiring more compute during the cosine similarity step. Table 1 shows the impact of top $x\%$ keywords on performance when tested on UCF-Crime.

Table 1: Effect of using top $x\%$ keywords on performance when tested on UCF-Crime.

| $x\%$ | 5 | 10 | 30 | 50 |
|---|---|---|---|---|
| Auc % | 87.09 | 87.21 | 87.22 | 87.19 |

The parameter $S$, representing the number of top-ranked words used, is determined by the maximum number of characters that can be accommodated within the CLIP text encoder's token limit of 77, following the sorting of words by their cosine similarity to class labels.

Training is carried out on an NVIDIA RTX 4090 GPU, using similar hyperparameter settings to original VADCLIP implementation. Specifically, we use a learning rate of $2 \times 10^{-5}$ for both the UCF-Crime and XD-Violence datasets. The batch size is set to 64 for UCF-Crime and 128 for XD-Violence, respectively.

## 4.4 Results

We evaluate the performance of our proposed method, SemVAD, against four recent state-of-the-art models—CLIP-TSA (Joo et al., 2023), PEL4VAD (Pu et al., 2024), UR-DMU(Zhou et al., 2023) and VAD-CLIP (Wu et al., 2024)—as well as the earlier baseline approach by Sultani et al. (Deep MIL) (Sultani et al., 2018). For a fair and robust comparison, we train each model using 10 different random seeds and report the mean performance along with the standard deviation.

Table 2: Coarse-grained performance comparison on UCF-Crime and XD-Violence datasets. Mean and standard deviation (SD), and the performance with the known best-performing seed are presented.

| | UCF-CRIME | | XD-VIOLENCE | |
|---|---|---|---|---|
| Method | Mean AUC(%) (SD) | Best Seed AUC(%) | Mean AP(%) (SD) | Best Seed AP (%) |
| RTFM (Tian et al., 2021) | N/A | 85.66 | N/A | 78.27 |
| S3R(Wu et al., 2022) | N/A | 85.99 | N/A | 80.47 |
| UMIL(Lv et al., 2023) | N/A | 86.75 | N/A | N/A |
| Deep MIL(Sultani et al., 2018) | 75.58 (0.77) | 77.92 | 73.24 (1.22) | 75.18 |
| UR-DMU(Zhou et al., 2023) | 85.52 (0.744) | 86.75 | 79.22 (1.206) | 82.41 |
| CLIP-TSA(Joo et al., 2023) | 85.02 (0.466) | 87.58 | 79.83 (1.05) | 82.17 |
| PEL4VAD(Pu et al., 2024) | 85.21 (0.366) | 86.76 | 84.21 (0.31) | 85.59 |
| VADCLIP(Wu et al., 2024) | 86.26 (0.918) | 88.02 | 83.20 (1.53) | 84.51 |
| **SemVAD** | **87.21 (0.462)** | **88.48** | **84.66 (1.26)** | **86.55** |

**Coarse-Grained Performance:** As shown in Table 2, SemVAD achieves the highest mean AUC among all methods evaluated, outperforming the state-of-the-art VADCLIP by 0.95% AUC while maintaining a relatively low standard deviation. This highlights both the effectiveness and stability of our approach. Notably, SemVAD also outperforms the reported best results of existing methods on both datasets. A similar trend is observed for the XD-Violence dataset, where SemVAD achieves the highest mean AP score of 84.66%, outperforming the most recent state-of-the-art method, PEL4VAD, by 0.45% and surpassing VADCLIP by 1.46%. It is also notable that, despite having a lower mean performance, PEL4VAD demonstrates the lowest standard deviation.

**Fine-grained Performance:** We conduct fine-grained evaluations using mean average precision (mAP) across five Intersection-over-Union (IoU) thresholds: 0.1, 0.2, 0.3, 0.4, and 0.5. We report the average mAP across these thresholds, referred to as mean average mAP, and the corresponding standard deviation across 10 different training seeds. Table 3 shows that SemVAD achieves an mean average mAP of 8.1%, outperforming VADCLIP by 1.15% when tested on UCF-CRIME. Similarly, a substantial improvement is observed with XD-Violence, with a mean average mAP of 31.615%, representing a 9.12% increase over VADCLIP. These results demonstrate that the semantic features of SemVAD help even more with fine-grained anomaly detection compared to coarse-grained anomaly detection.

## 4.5 Explainability

In Section 4.4, we report a significant improvement in fine-grained anomaly detection performance compared to recent state-of-the-art methods. This can be attributed to the fact that the generated captions $C_i$ encapsulate anomaly-specific information. Keywords such as *"shooting"*, *"fire"*, and *"pushing"*, among others, provide contextually relevant cues to the anomaly detector.

Figure 7(a–c) shows that captions $C_i$ effectively describe normal and anomalous segments, aiding interpretability. Figure 7(d) illustrates a misclassification— *"Burglary"*, *"Stealing"*, and *"Robbery"*—due to semantic overlap in captions (e.g., *"attempt"*, *"break"*), highlighting both the strengths and limitations of

Table 3: Fine-grained (multi-class) performance comparisons on UCF-CRIME and XD-VIOLENCE. * represents results without averaging over ten seeds as presented in (Wu et al., 2024).

| | | mAP IOU% | | | | | | |
|---|---|---|---|---|---|---|---|---|
| | Method | 0.1 | 0.2 | 0.3 | 0.4 | 0.5 | Mean Avg mAP % | SD |
| UCF-CRIME | Deep MIL* | 5.73 | 4.41 | 2.69 | 1.93 | 1.44 | 3.24 | N/A |
| | AVVD* | 10.27 | 7.01 | 6.25 | 3.42 | 3.29 | 6.05 | N/A |
| | VADCLIP | 12.32 | 8.91 | 6.25 | 4.29 | 2.99 | 6.95 | 1.44 |
| | **SemVAD** | 15.49 | 10.625 | 7.35 | 4.27 | 2.92 | **8.1** | **1.19** |
| XD-VIOLENCE | Deep MIL* | 22.72 | 15.57 | 9.98 | 6.20 | 3.78 | 11.65 | N/A |
| | AVVD* | 30.51 | 25.75 | 20.18 | 14.83 | 9.79 | 20.21 | N/A |
| | VADCLIP | 33.47 | 26.89 | 21.50 | 16.31 | 12.0 | 22.03 | 1.932 |
| | **SemVAD** | 44.41 | 36.95 | 29.91 | 24.34 | 19.07 | **31.15** | **1.29** |

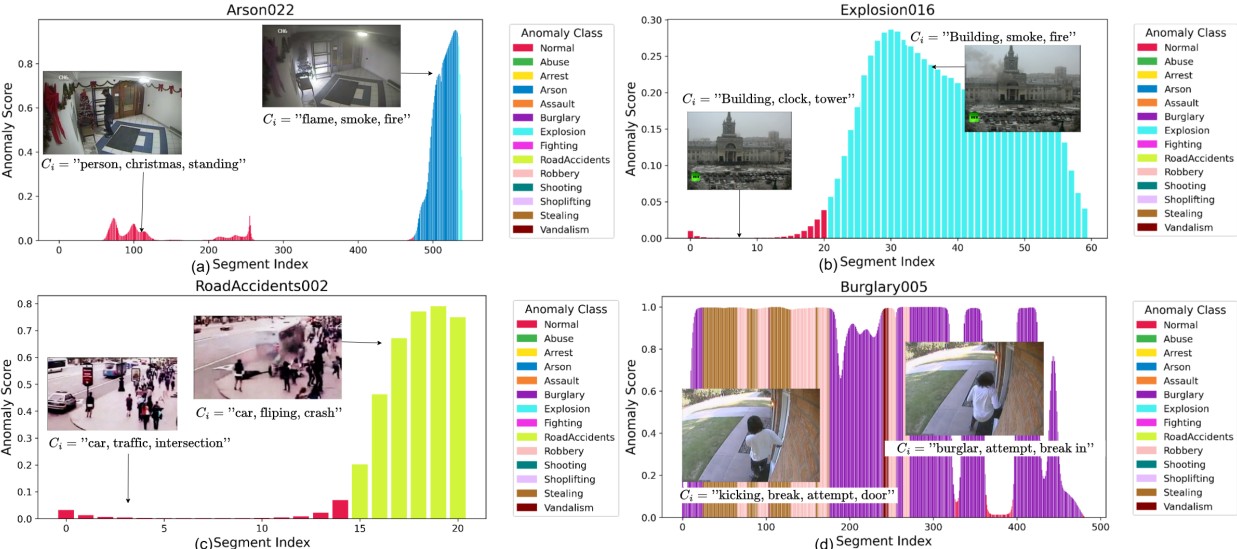

Figure 7: Explainability of fine-grained anomaly detection through VideoLLaMA 3 generated captions. Height denotes the coarse-grained anomaly score while the color indicates anomaly type.

caption-based reasoning. However, the distinction between *"Burglary"*, *"Stealing"*, and *"Robbery"* is naturally minimal.

To quantify the explainability performance, we manually annotated short clips from two videos per class in the UCF-Crime dataset, resulting in a total of 26 annotated videos. Captions generated by SemVAD for each video are then compared with the annotations using the cosine similarity score, yielding an average similarity score of 0.77. Table 4 shows some example scores together with the predicted captions human annotations. The annotations and predicted captions are given in the Appendix.

Table 4: VLM Generated Captions Compared to Human annotations

| Video id | VLM Generated | Cosine Similarity Score | Human Annotated Label |
|---|---|---|---|
| Robbery056 | A.2.1 | 0.92 | A.3.1 |
| Explosion021 | A.2.2 | 0.92 | A.3.2 |
| Robbery020 | A.2.3 | 0.42 | A.3.3 |
| Shoplifting006 | A.2.4 | 0.45 | A.3.4 |

### 4.6 Computational Cost

We also assess computational efficiency, as models like VideoLLaMA 3 are resource-intensive. Inference delay varies with caption length; we evaluated performance over 100 16-frame clips, passing through the entire SemVAD pipeline on an RTX 4090 (Table 5).

Table 5: Runtime analysis of SemVAD

| Mean time/clip (sec) | Standard deviation | Frames/clip | Estimated frames/second runtime |
|---|---|---|---|
| 1.126 | 0.243 | 16 | 14.2 |

We see that on average it takes approximately 1.126 seconds to generate an anomaly score for a given clip with a standard deviation of 0.243 seconds, which implies that the detector can process frames at a rate of 14.2 frames per second on average, which indicates it can be used real-time in many practical scenarios. We also provide a detailed runtime breakdown. Table 6 shows that the total inference per 16-frame clip is 1.126s on an RTX 4090. Of this, only 0.014s is due to our fusion + detection model, while 1.112s stems from VideoLLaMA-3 caption generation. The overhead therefore comes almost entirely from captioning.

Table 6: Module-wise runtime analysis (per 16-frame clip).

| Module | Time (s) | Fraction |
|---|---|---|
| VideoLLaMA-3 captioning | 1.112s | 98.8% |
| Fusion + detection model | 0.014s | 1.2% |
| **Total** | **1.126** | 100% |

To contextualize the cost of captioning, we compare it with prior approaches that rely on both traditional (RGB + optical flow features from I3D or CLIP visual encoders) and new VLM-based approaches in Table 7. Extracting I3D RGB+Flow features typically requires 20ms per clip on similar GPUs, while CLIP visual encoding is much faster (15ms/clip). Our captioning cost (1.112s) is thus substantially heavier than CLIP and I3D. However, unlike I3D+flow and CLIP, captions provide semantic interpretability. Recently, there has been growing interest in using VLMs and LLMs to generate anomaly scores without training, e.g., LAVAD Zanella et al. (2024) which uses a VLM (BLIP2) and an LLM (Llama-2) together to generate anomaly scores for videos. As seen in Table 7, LAVAD takes 3.9 seconds, much more than captioning in SemVAD with VideoLLaMa-3.

Table 7: Runtime comparison across models (per 16-frame clip)

| Model | Time (s) | Task |
|---|---|---|
| I3D RGB + Flow (two-stream) | 20ms | CNN-based vision features |
| CLIP visual encoder | 15ms | Transformer-based vision features |
| SemVAD fusion + detection | 14ms | Vision + caption fusion and detection |
| VideoLLaMA-3 captions | 1.112s | VLM-based semantic captions |
| LAVAD | 3.9s | Training-free VLM+LLM for detection |

presents a comparative evaluation of Video-LLaMA-3-7B and its lightweight counterpart, Video-LLaMA-3-2B. When integrated into the trained SemVAD framework, the 2B model exhibits a modest reduction of 0.92% and a 0.74% in coarse grained and fine grained performance respectively relative to the 7B variant, while achieving a substantial efficiency gain of approximately 392 ms per inference at a substantially lower memory cost. This improvement enables real-time operation at 22 fps, compared to 14 fps with the larger model. These findings highlight that caption generation constitutes the primary computational bottleneck, thereby underscoring the value of lightweight captioning models for deployment in resource-constrained settings. Nevertheless, the rapid advancement of hardware and the growing adoption of large language models (LLMs) in this domain suggest that the use of larger models may soon become practically feasible.

Table 8: Compute speed and memory usage comparison between Video-llama-3-7b and Video-llama-3-2b

| | UCF-CRIME Coarse-grained | UCF-CRIME Fine-grained | Time (s) | GPU Memory |
|---|---|---|---|---|
| VideoLlama-3-2b | 86.34 | 6.68 | 0.72 | 5.6 GB |
| VideoLlama-3-7b | **87.26** | **7.42** | 1.112 | 17.3 GB |

## 4.7 Transferability

In this section, we evaluate the transferability of models trained on distinct datasets. UCF-Crime represents a real-world anomaly detection dataset consisting primarily of low-resolution, static surveillance footage across a wide variety of scenarios. In contrast, XD-Violence is composed largely of clips from movies and films, which typically feature higher-resolution, dynamic, and non-static scenes. Due to these substantial differences in data characteristics, models trained on one dataset generally exhibit limited transferability to the other.

However, SemVAD aims to mitigate this gap by incorporating semantic captions, thereby leveraging semantic similarities across datasets to enhance generalization. As shown in Table 9, SemVAD achieves superior

Table 9: Transferability analysis between SemVAD and VADCLIP.

| | Training Data UCF-Crime | Testing Data XD-Violence | Training Data XD-Violence | Testing Data UCF-Crime |
|---|---|---|---|---|
| SemVAD | AP: 71.66% | | AUC: 84.66% | |
| VADCLIP | AP: 66.66% | | AUC: 81.1% | |

transferability compared to the baseline VADCLIP, highlighting the effectiveness of incorporating semantic information from video captions in bridging the domain gap between heterogeneous datasets.

## 4.8 Ablation Study

In addition to incorporating video captions $C_i$ into the model, we propose three key modules: an attention fusion mechanism, a convolutional mixer, and a contrastive class-specific loss ($\ell_{\text{sem}}$). To evaluate the contribution of each component, we conduct a comprehensive ablation study. Table 10 presents the impact of each module on the mean AUC for UCF-Crime, computed over 10 different random seeds. We observe that directly concatenating the caption features $T_i$ with the CLIP vision-encoded features $F_i$ yields a 0.25% improvement over the VADCLIP baseline. Introducing the attention fusion module further enhances performance by 0.142%. The addition of the convolutional mixer results in a substantial gain of 0.438%, underscoring its significant contribution to overall model effectiveness. Finally, incorporating the $\ell_{\text{sem}}$ loss provides an additional performance increase of 0.132%, leading to the highest mean AUC of 87. 212%.

Table 10: Ablation study showing the contribution of each proposed component on UCF-Crime. Performance is reported as the mean AUC (%) over 10 random seeds.

| Modules | | | | Mean ROC AUC (%) | Performance Increase (%) |
|---|---|---|---|---|---|
| Caption ($C_i$) | Attention Fusion | Convolution Mixer | $\ell_{\text{sem}}$ | | |
| ✓ | | | | 86.50 | +0.25 |
| ✓ | ✓ | | | 86.642 | +0.142 |
| ✓ | ✓ | ✓ | | 87.08 | **+0.438** |
| ✓ | ✓ | ✓ | ✓ | **87.212** | +0.132 |

### 4.8.1 Variance in captions

Our model generates video captions using greedy decoding, which produces deterministic captions for each video. To investigate the impact of caption variability on performance, we conduct experiments with increased randomness by adjusting the temperature and *top_p* parameters.

Table 11: Impact of caption variability on performance.

|  | UCFCRIME Coarse-grained AUC | UCFCRIME Fine-grained mAP |
|---|---|---|
| Temperature = $top\_p = 0.3$ | 86.2 | 5.48 |
| Temperature = $top\_p = 0.7$ | 85.85 | 4.99 |
| Greedy Decoding | 87.26 | 8.1 |

As shown in Table 11, introducing higher randomness through elevated temperature and *top_p* values results in a noticeable reduction in performance compared to greedy decoding, indicating that deterministic captions are more effective for this task.

## 5 Limitations and Future Directions

The use of a VLM for streaming video may cause some computational burden, especially for resource-constrained systems. Specifically, VideoLLaMA 3, which is used in the experiments, requires a GPU. Hence, the proposed method is not suitable for a standard computer or an edge device, and more efficient methods for resource-constrained systems can be a promising research direction. However, as discussed in Section 4.5, the proposed method can run near real-time (14.2 fps) on a reasonable computer with and NVIDIA RTX 4090 GPU.

In Section 4.2, we address the issue of stochastic variability affecting MIL-based methods, identifying two primary contributing factors: the random pairing of anomalous and nominal videos within each training batch and the random initialization of model weights at the onset of training. While a comprehensive investigation into enhancing the robustness of such models warrants dedicated research, one potential approach to mitigating this variability is to limit stochasticity during the early phases of training.

Although it is difficult to fully control the random pairing of normal and anomalous segments, a feasible alternative is to select the top-$k$ segments based on their cosine similarity to nominal features at the beginning of training. This method can potentially mitigate the instability introduced by random weight initialization.

To further address this issue, we incorporate a $\sigma$-greedy strategy, commonly used in reinforcement learning. For each training batch comprising $b$ samples—consisting of an equal number of anomalous and normal instances (i.e., $b/2$ each)—we introduce a control parameter $\sigma$, initialized to 1. At each training iteration, $\sigma$ is updated according to the exponential decay rule:

$$\sigma_{new} = 0.99 * \sigma.$$

A random value $\alpha$ is then sampled uniformly from the interval $[0, 1]$. If $\alpha < \sigma$, the top-$k$ anomalous segments in each anomaly-containing video are selected based on the average cosine distance between the segment and all other nominal segments within the batch. Segments exhibiting the highest cosine distances are considered anomalous. Conversely, if $\alpha \geq \sigma$, segment selection reverts to the standard MIL procedure, where segments are chosen based on the model's classification scores.

This approach allows the model to rely on cosine similarity during the early stages of training, when model predictions are unreliable, and gradually transitions to using its own predictions as training progresses.

Table 12 presents a performance comparison between the $\sigma$-greedy approach and standard MIL training on both SemVAD and the vanilla VADCLIP framework using the UCF-CRIME dataset. The results indicate that the $\sigma$-greedy strategy contributes to more stable training, as evidenced by a reduction in standard deviation, albeit with a slight trade-off in the mean performance. Although the pairing of anomalous and nominal videos within each batch remains stochastic, leveraging cosine similarity as a selection criterion—while not always optimal—offers additional stability during the training process.

Table 12: Performance comparison between SemVAD and VADCLIP when trained using the $\sigma$-greedy method.

|  | SemVAD Mean AUC (%) (SD) | VADCLIP Mean AUC (%) (SD) |
|---|---|---|
| $\sigma$-greedy | 87.08 (0.248) | 86.12 (0.694) |
| Regular | 87.21 (0.462) | 86.26 (0.917) |

## 6 Conclusion

In this work, we propose SemVAD, a novel framework for weakly supervised video anomaly detection (wVAD) that integrates semantically rich captions generated by a multimodal language model with visual representations extracted from CLIP. By introducing a two-phase fusion architecture, comprising attention-based feature alignment and a convolutional mixer—along with a class-specific semantic alignment loss, we demonstrate significant performance gains with a reasonable computational cost.

Our experimental results establish SemVAD as a new state-of-the-art in wVAD. Specifically, SemVAD consistently surpasses recent benchmarks such as VADCLIP and PEL4VAD, achieving higher coarse-grained detection accuracy and substantially improving fine-grained anomaly localization. The integration of semantic captions contributes not only to enhanced performance but also to explainability, as natural language descriptions provide interpretable cues about detected anomalies. Furthermore, our transferability study highlights that SemVAD is also better equipped to generalize across data domains, as we observe increased transferability of learned knowledge across two distinct datasets.

The common practice of evaluating performance using the "best" known seeds for random number generators is also criticized. We showed that due to the underlying characteristics of MIL-based training, there is a wild fluctuation in the performance of state-of-the-art methods as a function of randomness seed. For a more reliable comparison and more reproducible results, we propose to evaluate the wVAD methods in terms of mean and standard deviation of performance over a number of runs with different seeds.

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

# A Appendix

## A.1 System Prompt

"Look at the video carefully. Describe the activity is abnormal or criminal (e.g., fighting, robbery, burglary, vandalism, assault, road accident, shooting, abuse, or other unusual violent actions). If no anomaly is present, describe the scene as normal daily activity (e.g., walking, standing, shopping, driving). Keep your description short."

### A.2  VLM Responses

### A.2.1

Two men are robbing a store while the security guard tries to stop them. One of the men pulls out a gun and points it at the guard, demanding money. The guard complies and hands over the cash.

### A.2.2

A person is filming an explosion at a gas station. The explosion occurs and the building is destroyed.

### A.2.3

In the video, a man wearing a yellow shirt is standing in front of an ATM machine and using his cell phone. Another man wearing a brown coat enters the scene and sits on the floor next to him. The man in the brown coat then grabs the yellow-shirted man's collar and pulls out a gun, pointing it at him. The yellow-shirted man appears to be complying with the demands of the man with the gun.

### A.2.4

A group of people are gathered around a man who is sitting on a chair. They are talking to him and he is holding a cup in his hand. The man is wearing a blue hat. There is also a woman standing nearby.

### A.3  Human Annotations

### A.3.1

Two armed men rob a store while a security guard tries to intervene.

### A.3.2

An explosion destroys a gas station.

### A.3.3

A man at an ATM is threatened by another man with a gun during a robbery.

### A.3.4

A man in a jewelry store grabs a necklace and runs out of the store.

### A.4  QKV Attention instead of QKK

In Section 3.3.1, we adopt a QKK attention mechanism in place of the conventional QKV formulation. **Empirical evidence:** When testing with **UCF-Crime**, we observed higher performance on both coarse grained and fine-grained performance using QKK. We expand this by including results with **XD-Violence**, where QKV obtains AP = 83.13 and fine-grained mAP = 31.57. These results confirm that QKK is competitive with or slightly superior to QKV. Table 13 shows the comparison between the two approaches. QKK approach surpasses QKV on both coarse grained metrics except fine-grained performance on XD-Violence that is slightly higher with QKV.

Table 13: QKV vs QKK Comparison

|  | UCF-CRIME Coarse-grained | UCF-CRIME Fine-grained | XD-Violence Coarse-grained | XD-Violence Fine-grained |
|---|---|---|---|---|
| QKV | 86.91 | 8.05 | 83.13 | **31.57** |
| QKK | **87.21** | **8.13** | **84.66** | 31.15 |

**Theoretical rationale:** In cross-modal fusion, the caption embeddings are already semantically aligned (VideoLLaMA $\rightarrow$ CLIP text space). Using $t_i$ as both $V$ and $K$ avoids redundancy with a smaller model (without additional parameters for $V$ projection), which helps with regularizing the model for better generalization performance.

### A.5 VAD Clip loss

**Binary Cross-Entropy Loss ($L_{bce}$).** This loss supervises the coarse-grained anomaly detection branch by comparing the predicted video-level anomaly score with the video label:

$$L_{bce} = -\big[y\log(p) + (1-y)\log(1-p)\big], \tag{2}$$

where $y \in \{0, 1\}$ is the video-level label (0 for normal and 1 for abnormal), and $p$ is the predicted video-level anomaly score obtained by aggregating the top-$K$ frame-level anomaly scores. This loss encourages correct discrimination between normal and anomalous videos.

**Noise-Contrastive / Alignment Loss ($L_{nce}$).** This loss aligns video features with the correct anomaly text embedding:

$$p_i = \frac{\exp(s_m/\tau)}{\sum_j \exp(s_j/\tau)}, \quad L_{nce} = -\log(p_y), \tag{3}$$

where $s_m$ is the similarity between the video representation and the $m$-th text class embedding, $\tau$ is a temperature parameter, $p_m$ is the predicted probability of class $m$, and $y$ is the ground-truth anomaly class index, while $j$ are all classes other than $m$. This loss enforces high similarity between the video and its correct textual anomaly category.

**Contrastive Text Separation Loss ($L_{cts}$).** This loss enforces separation between the normal class embedding and abnormal class embeddings:

$$L_{cts} = \sum_{m=1}^{C} \max\big(0, \cos(t'_{p=0}, t'_{p=m})\big), \tag{4}$$

where $t'_{p=0}$ is the text embedding of the normal class, $t'_{p=m}$ is the embedding of the $m$-th abnormal class, $\cos(\cdot, \cdot)$ denotes cosine similarity, and $C$ is the number of abnormal classes. This loss pushes normal and abnormal semantic representations apart in the embedding space.

**Overall Objective.** The final training objective is given by:

$$L = L_{bce} + L_{nce} + \lambda L_{cts}, \tag{5}$$

where $\lambda$ is a weighting factor that controls the contribution of the contrastive text separation loss. The values for $\tau$ and $\lambda$ are set to the defaults form Wu et al. (2024).

### A.6 Broader Impacts

The improved performance and explainability of proposed VAD method can benefit the society in public safety (through surveillance of public areas) and internet safety (through moderating content posted on the internet). Crimes or dangerous activities in public areas can be automatically detected for alerting public safety professionals. Similarly, videos with harmful content can be detected before being published on the internet. The explainability feature of the proposed method can help professionals to trust the video AI more. Nevertheless, like all video surveillance technologies, the proposed VAD method should be used in an ethical way , particularly in terms of data privacy, fairness, and responsible use.

