# OpenReview forum: "SemVAD: Fusing Semantic and Vision Features for Weakly Supervised Video Anomaly Detection"
_TMLR — Accepted by TMLR_

### Review · Reviewer_NT1F · 2025-09-30

**Summary Of Contributions:**

Summary:
The paper introduces SemVAD, a framework for weakly supervised video anomaly detection (wVAD) that integrates semantic information from captions generated by VideoLLaMA 3 with visual features from CLIP. The authors also emphasize the instability of MIL-based methods under random initialization and propose evaluating models over multiple random seeds. Experiments on UCF-Crime and XD-Violence show that SemVAD achieves SOTA results in anomaly detection.

Strength:
(1)The paper highlights seed sensitivity in MIL-based wVAD and systematically addresses it by reporting mean/std over multiple runs. This sets a good precedent for future work.
(2)The method consistently outperforms state-of-the-art baselines on both UCF-Crime and XD-Violence, in both coarse- and fine-grained detection.
(3)Each proposed component is justified experimentally (captions, attention, convolution mixer, semantic loss).

Weakness：
(1)The choice of QKK attention instead of QKV is unconventional and only lightly justified in the appendix. A deeper theoretical or empirical motivation would be beneficial.
(2)Although runtime is measured, reliance on VideoLLaMA 3 introduces significant overhead. The method may be impractical for real-world low-resource deployments, and efficiency trade-offs are not deeply analyzed.
(3)Some figures (e.g., Fig. 2, Fig. 4) use symbols such as Fᵢ and Lᵥ without clear captions or explanations, which reduces readability. The authors should clarify these symbols in the figure captions or within the diagrams.
(4)The efficiency analysis lacks a module-wise runtime breakdown and comparison with baselines.

**Audience:**

Yes

**Audience Explanation:**

The findings would interest TMLR’s audience since weakly supervised video anomaly detection is an important problem in computer vision with practical relevance to surveillance and content moderation.

**Broader Impact Concerns:**

I have no ethical concerns regarding this work.

**Claims And Evidence:**

Yes

**Claims Explanation:**

Most claims are supported with convincing evidence through solid results on UCF-Crime and XD-Violence, multi-seed robustness analysis, and ablations. However, support for efficiency and explainability is weaker, as runtime breakdowns and quantitative explainability metrics are missing.

**Requested Changes:**

(1) Provide a stronger justification for adopting QKK attention instead of the conventional QKV, either through additional ablation experiments or a deeper theoretical explanation.

(2) Expand the efficiency analysis by discussing the computational overhead of VideoLLaMA 3 and exploring potential trade-offs or lightweight alternatives to demonstrate feasibility for low-resource settings.

(3) Improve the readability of figures (e.g., Fig. 2, Fig. 4) by adding clear captions or annotations for symbols such as Fᵢ and Lᵥ.

(4) Provide a module-wise runtime analysis (caption generation vs. fusion vs. detection) and compare with baseline methods to support claims of efficiency and practicality.

---

### Review · Reviewer_qf3a · 2025-10-08

**Summary Of Contributions:**

This paper proposes SemVAD, a novel framework for weakly supervised video anomaly detection (wVAD). It fuses semantic captions generated from a video-language model (VideoLLaMA 3) with visual embeddings from the CLIP model. Authors innovated a new fusion mechanism via a two-phase module consisting of an attention-based vision caption feature fusion and a convolutional mixer. A class-specific semantic alignment loss is added to improve fine-grained discrimination. Further, the authors address randomness sensitivity in multiple-instance learning by averaging across seeds. Their experiments on UCF-Crime and XD-Violence show that SemVAD outperforms SotA methods such as VADCLIP and PEL4VAD on both coarse- and fine-grained tasks.

Strengths:
1. Innovative integration of VLM-generated semantics into wVAD, enhancing interpretability
2. Systematic evaluation across multiple seeds for reproducibility
3. Clear empirical gains (≈ +0.9 AUC and +1.46 AP) and meaningful improvements in fine-grained anomaly localization
4. Well-documented ablation and transferability analyses

Weaknesses:
1. Computational overhead.
The reliance on a heavy VLM (VideoLLaMA 3) restricts deployment to high-end GPUs and prevents use on edge or embedded systems. Even with an RTX 4090, processing speed (~14.2 fps) leaves little margin for scaling to multi-camera or higher-resolution settings.
2. Empirical but weakly justified design.
The two-phase fusion module, comprising attention and conv-mixer, and the semantic-alignment loss are empirically validated but lack strong theoretical grounding. Choices like the QKK attention variant (A.1) are motivated mainly by incremental gains rather than principled reasoning about modality alignment.
3. Semantic noise sensitivity.
Since captions drive anomaly localization, errors or overlaps in textual descriptions (e.g., burglary, stealing, and robbery) can mislead the model. The heuristic prompt-selection pipeline may amplify dataset biases or truncate informative words, reducing robustness.

**Audience:**

Yes

**Audience Explanation:**

The topic of leveraging multimodal foundation models for weakly supervised video anomaly detection lies closely within the intersection of machine learning, computer vision, and multimodal representation learning, which aligns well with TMLR’s audience. The focus on reproducibility and interpretability via multi-seed evaluation and the integration of large VLMs will be of interest to researchers in both applied ML and those analyzing robustness in weak supervision.

**Broader Impact Concerns:**

The paper includes a brief discussion of Broader Impacts in Appendix A.2, noting potential societal benefits, such as improved public and internet safety through automated surveillance and content moderation. These points are appropriate and align with the intended application domain. However, it could be slightly strengthened by briefly reiterating the importance of ethical deployment, particularly in terms of data privacy, fairness, and responsible use.

**Claims And Evidence:**

Yes

**Claims Explanation:**

The empirical results substantiate the paper’s claims of improved detection accuracy, robustness, and explainability. The experiments follow fair comparisons and show statistically consistent gains. Ablation and transfer experiments reinforce the causal impact of each module. While computational efficiency is only partially addressed, the quantitative and qualitative evidence support the main contributions convincingly.

**Requested Changes:**

1. Clarify fusion design:
Provide conceptual or mathematical intuition for why the two-phase (attention + conv mixer) fusion outperforms simpler concatenation or transformer-based fusion.

2. Quantify explainability gains:
Beyond visual examples, include a human-evaluated or textual-similarity metric quantifying how captions improve interpretability.

3. Address caption quality variance:
Discuss how caption noise from VideoLLaMA affects anomaly scoring, perhaps with an uncertainty or filtering mechanism.

4. Expand computational feasibility discussion:
Report GPU memory usage and runtime under different model scales; Suggest lighter VLM alternatives.

5. Minor suggestions:
Ensure consistent use of seed-averaged metrics throughout tables; Fix minor typos (e.g., “comprison” in Table 7 caption).

---

### Review · Reviewer_og3b · 2026-01-08

**Summary Of Contributions:**

This paper proposes a new algorithm for anomaly video detection. Comparing to [Wu et. al, 2024], this paper introduces and fuses the visual embeddings with the caption embeddings for the clips in a video. Moreover, based on the clip captions, for each anomaly class $m$, it computes the embedding $T^m$ using the most important keywords extracted from the clips belonging to this anomaly class. The experimental results show that the proposed approach outperforms the current SOTA ones.

**Audience:**

Yes

**Audience Explanation:**

The video anomaly detection is an important application in video processing and analyzing.

The way this paper utilizes the clip captions is reasonable and interesting.

**Broader Impact Concerns:**

No.

**Claims And Evidence:**

Yes

**Claims Explanation:**

The results and analysis have demonstrated the role of clip (video) captions in improving the performance because the captions contain relevant keywords for video anomaly detection.

The workaround to get the anomaly class embedding $T^m$ from all clip captions belonging to this class is reasonable and interesting.

**Requested Changes:**

Figure 4 and Section 3.3.1 are not technically solid and need changes. According to the current writing, $f_i$ and $t_i$ are two D-dimensional vectors and it does not make sense to do cross-attention between them. Instead, they should be 2D tensors of visual embeddings and caption embeddings of the clips in a video.

The loss $\ell_{sem}$ needs more clarification. For example, the purpose of this loss is to encourage the feature extractor and classification head to predict top-K most relevant clip regarding an anomaly class $m$ so that their visual embeddings well align to the anomaly class embedding $T^m$.

Moreover, the losses reused from VADCLIP also need more detailing introduction at lease in the appendix for the self-containing of the paper.

---

> ### Author Response · Authors · 2026-01-13
> **Requested Changes.**
>
> We thank the reviewer for their insightful comments. We have addressed all the requested changes as detailed below:
>
> 1. Figure 4 has been updated, and the corresponding description in Section 3.3.1 (highlighted in the manuscript) has been revised to accurately reflect the attention fusion process and its correct dimensionality.
>
> 2. The description of the $L_{\text{sem}}$ loss in Section 3.4 has been revised to more precisely capture the purpose and formulation  of the loss function. The updated text is highlighted for clarity.
>
> 3. The appendix has been expanded with a new Section A.5, which details the loss functions previously used by VADCLIP, as  requested.

---

> > ### Comment · Reviewer_og3b · 2026-02-07
> > **Thanks for your update**
> >
> > The authors have updated the paper accordingly. My concerns have been addressed.

---

### Decision · Action_Editor_HyqJ · 2026-02-15

**Recommendation:** Accept as is

**Additional Comments:**

Two reviews "lean accept" while one rates "lean reject".  The reasons to lean reject were a concern that the method's complexity is not justified by commensurate improvements.  However, as all of the reviewers agree that the paper offers original contributions that are well evaluated and of interest to the community, the paper meets the acceptance criteria.

**Audience:**

Yes

**Audience Explanation:**

The topic of video anomaly detection is of sufficient interest, and the reviewers all agree that at least some individuals in the TMLR audience would be interested.

**Claims And Evidence:**

Yes

**Claims Explanation:**

The reviewers agree that the paper offers thorough experimentation and ablations to evaluate the contributions and design decisions. In one review, additional analysis was suggested, and the authors updated to include it.